# Oral Intake of Chicken Bone Collagen Peptides Anti-Skin Aging in Mice by Regulating Collagen Degradation and Synthesis, Inhibiting Inflammation and Activating Lysosomes

**DOI:** 10.3390/nu14081622

**Published:** 2022-04-13

**Authors:** Changwei Cao, Zhichao Xiao, Huiquan Tong, Yuntao Liu, Yinglong Wu, Changrong Ge

**Affiliations:** 1College of Food Science, Sichuan Agricultural University, Ya’an 625014, China; ccwylf1111@163.com (C.C.); liuyt@sicau.edu.cn (Y.L.); 2Livestock Product Processing Engineering and Technology Research Center of Yunnan Province, Yunnan Agricultural University, Kunming 650201, China; ynzcxiao@126.com; 3Graduate Department, Kunming University, Kunming 650214, China; tonghuiquan1988@163.com

**Keywords:** collagen peptides, anti-aging, skin transcriptome, collagen synthesis, lysosomes

## Abstract

The effect of diet on skin aging has become an interesting research topic. Previous studies have mostly focused on the beneficial effects of collagen peptides derived from marine organisms on the aging skin when administered orally, while the beneficial effects of collagen peptides derived from poultry on aging skin have been rarely reported. In this study, collagen peptides were prepared from chicken bone by enzymatic hydrolysis, and the effect and mechanism of action of orally administered collagen peptides on alleviating skin aging induced by UV combined with D-galactose were investigated. The results showed that the chicken bone collagen had typical characteristics of collagen, and the chicken bone collagen peptides (CPs) were mainly small molecular peptides with a molecular weight of <3000 Da. In vivo experiments showed that CPs had a significant relieving effect on aging skin, indicated by the changes in the compostion and structure of the aging skin, improvement of skin antioxidant level, and inhibition of inflammation; the relieving effect was positively correlated with the dose of CPs. Further investigation showed that CPs first reduce the level of skin oxidation, inhibit the expression of the key transcription factor AP-1 (c-Jun and c-Fos), then activate the TGF-β/Smad signaling pathway to promote collagen synthesis, inhibit the expression of MMP-1/3 to inhibit collagen degradation, and inhibit skin inflammation to alleviate skin aging in mice. Moreover, the skin transcriptome found that lysosomes activated after oral administration of CPs may be an important pathway for CPs in anti-skin aging, and is worthy of further research. These results suggested that CPs might be used as a functional anti-aging nutritional component.

## 1. Introduction

Health era, accurately following a healthy diet, determining the role of nutrition in skin aging, and deciding what to eat to maintain youthful and healthy skin are difficult problems. Skin is the largest organ of the body, composed of the epidermis, dermis, and subcutaneous layer. It acts as a barrier to protect the body from external factors and also plays a role in health and beauty [1]. Skin aging is a complex process, which is divided into chronological aging and photo-aging and is affected by both internal and external factors. The main characteristics include the accumulation of macromolecular damage in the cell, the decline of stem cell function (promoting tissue renewal), and the gradual loss of the physical integrity of the skin [2]. The main molecular mechanisms that cause skin aging mainly include oxidative stress, telomere shortening, DNA damage, genetic mutation, micro-RNA regulation, advanced glycation end-product accumulation, and inflamm-aging [3]. Photo-aging is skin epidermal hyperplasia, drying, and extracellular matrix degradation induced by ultraviolet radiation. The main cause of photo-aging is reactive oxygen species (ROS) generated by ultraviolet radiation that mediate the expression of matrix metalloproteinases (MMPs) and type-I pro-collagen through signal pathways such as the mitogen-activated protein kinase (MAPK) signaling pathway, thus leading to the degradation of the extracellular matrix (ECM) in the skin and the apoptosis of fibroblasts [4]. In recent years, maintaining skin health and delaying aging has become popular, and finding natural ingredients such as bioactive peptides and polyphenols with antioxidant and anti-aging functions has become a research hotspot [5].

However, collagen has a large molecular weight and is difficult to be directly absorbed and utilized, whereas the small molecular collagen peptides after collagen hydrolysis have stronger biological activity and a high absorption rate [6]. Meanwhile, the wide application of collagen in food, medicine, tissue engineering, cosmetics, and other fields has made collagen peptides with lower molecular weight, higher absorption efficiency, and stronger biological activity a new favorite in functional food and medical research [7,8,9]. Collagen peptides are small, containing 2–20 amino acid residues obtained after collagen hydrolysis. They are used as a dietary supplement for the treatment of skin aging due to their potential anti-inflammatory and antioxidant functions and immune regulation and the effects of antioxidation and proliferation on skin fibroblasts [10].

After oral administration, collagen peptides are absorbed in the form of small peptides and quickly transported to the blood, and eventually, to the kidneys, skin, joints, and other tissues to be stored and used. After 14 days, radioactivity levels remained high in the skin of mice that had been gavaged with C^14^-labeled collagen peptides. The collagen peptides can be almost completely absorbed and utilized by the body, whereas the absorption and utilization rate of collagen is only 50–60% [6,11,12,13]. In recent years, collagen hydrolysates from fish skin, fish scale, cow bone, cow skin, and pig skin have been widely reported to have beneficial effects on alleviating skin aging and have received considerable attention from researchers. For example, collagen peptides isolated from silver carp skin promote pro-collagen synthesis and inhibit AP-1, MMP-1, and MMP-3 protein expression by activating the TGF-β/Smad pathway to prevent collagen degradation and repair photoaged skin cells [14]. Oral administration of bovine collagen peptides can improve skin relaxation, increase collagen content and antioxidant enzyme activity, repair collagen fiber, and normalize the collagen ratio of the skin in aging mice [15]. However, collagen peptides from different sources have different effects. The amount and structure of highly active peptides in the blood after oral administration of collagen peptides depend on the source of collagen [10]. The protective effect of collagen peptide extracted from the skin of hens against UVA-induced fibroblast injury was superior to that of the collagen peptide extracted from pig, cow, or tilapia skin, and its effect was equivalent to collagen-derived tripeptide (Gly-Pro-Hyp) [16].

Religious beliefs and disease concerns (such as mad cow disease) have led people to gradually shift the direction of the development of collagen and its products from terrestrial mammals to poultry and marine organisms [17]. As the main by-product of poultry processing, chicken bone is a promising source of collagen products. It not only reduces wastage of resources and environmental pollution but also allows the efficient utilization of by-products. Therefore, in this study, we used chicken bone collagen peptides as raw materials, with nude mice treated with D-galactose and UV to induce skin aging, as the model to evaluate the effect of oral administration of the collagen peptide on alleviating skin aging in mice and determine the relevant mechanism.

## 2. Materials and Methods

### 2.1. Materials, Chemicals, and Animals

The chicken farms of Yunnan Agricultural University (Kunming, China) provided the spent hens, which were separated, dried, and crushed to obtain the chicken bone meal. Superoxide dismutase (SOD), catalase (CAT), and glutathione peroxidase (GSH-PX) kits were purchased from Soleibao Biotechnology Co., Ltd. (Beijing, China). Hydroxyproline (HYP), interleukin-1α (IL-1α), matrix metalloproteinase-1/3 (MMP-1/3), type I pro-collagen, and hyaluronic acid (HA) enzyme-linked immunoassay (ELISA) kits were purchased from Nanjing Jiancheng Bioengineering Institute (Nanjing, China). Healthy female BALB/C hairless mice (18 ± 2 g, six weeks old) were purchased from Nanjing Junke Bioengineering Corporation, Ltd. (Nanjing, China) with permit number: SCXK(SU)2016–0010. Pepsin and papain were purchased from Aladdin Biochemical Technology Co., Ltd. (Shanghai, China), and other chemicals were of analytical grade. All mice were handled following the Regulations of the Laboratory Animal Care of Yunnan Province and approved by by the Yunnan Agricultural University Animal Care and Use Committee (approval ID:YAUACUC01).

### 2.2. Collagen Preparation and Amino Acid Composition

The extraction and amino acid composition of chicken bone collagen followed the method described by Liu et al. [18] with slight modifications. The chicken bone powder was stirred and soaked in 0.05 M NaOH, 10% n-hexane, and 0.25 M EDTA solution (pH 7.5) at a ratio of 1/10 (*m*/*v*). At each step, the sample was treated for 36 h, and the soaking solution was changed every 6 h to swell the bone powder, remove non-collagen proteins, fat, and minerals from the bone powder. The sample was washed with pure water to neutrality after each step. The pre-treated chicken bone meal was mixed with 0.5 M glacial acetic acid containing 0.1% (*w*/*v*) pepsin at a solid-to-liquid ratio of 1:10 (*w*/*v*) and continuously stirred and extracted at 4 °C for 48 h. It was then filtered, and the filtrate was centrifuged at 15,000× *g* for 15 min at 4 °C. The pH of the supernatant was adjusted to 7.5–7.8 with NaOH solution, and NaCl was added to a final concentration of 1.5 M. After the mixture was kept undisturbed for 12 h at 4 °C for salting out, the collagen precipitate was centrifuged at 15,000× *g* for 15 min at 4 °C, then dissolved with 0.5 M acetic acid, dialyzed in pure water, freeze-dried, and the finished product was chicken bone collagen.

The amino acid composition of chicken bone collagen was determined by the Sykam S433D amino acid automatic analyzer (Munich, Germany). A certain amount of collagen sample to be tested was taken in a sealed tube, added with 10 mL 6 M HCl, and hydrolyzed at 110 °C for 24 h. The hydrolysate was concentrated by nitrogen blowing and re-dissolved in 20 mL citric acid buffer. After microfiltration with a 0.22 μm microporous membrane, 20 µL hydrolysate was taken for amino acid spectrum analysis. The amino acid content in the sample was expressed in percentage.

### 2.3. Preparation and Molecular Weight Distribution of Collagen Peptides (CPs)

According to the results of our previous optimization process, CPs were prepared using papain at an enzyme-to-substrate ratio of 1:40 (mass ratio). After adjusting the pH to 7 and enzymatic hydrolysis at 63 °C for 5 h, the enzyme was inactivated in a boiling water bath for 15 min. The enzyme hydrolysate was desalted and lyophilized, re-dissolved in 0.1% formic acid solution, and centrifuged to obtain the supernatant. A Q Exactive HF Orbitrap High-Resolution Mass Spectrometer-QE-HF (Thermo Fisher, Waltham, MA, USA), equipped with electrospray ionization (ESI), was used to analyze the collagen hydrolysate in the full scan mode of 350–1800 m/z, and the results were retrieved and analyzed using the Proteome Discoverer 2.1 software.

### 2.4. Animal Test

Female BALB/C hairless mice (*n* = 55) were kept in a room under controlled conditions of temperature (24 ± 1 °C), humidity (60 ± 5%), and a 12 h day-night cycle for one week. They were provided ad libitum access to food and water. After one week of adaptation, the mice were randomly divided into the following five groups (*n* = 11 per group):(i).Normal group (N): UV unexposed; oral administration of 0.4 mL saline daily.(ii).Model group (M): UV exposed + D-galactose (0.2 mL); oral administration of 0.4 mL saline daily.(iii).Low dose collagen peptides group (LCPs): UV exposed + D-galactose (0.2 mL); oral administration of 0.4 mL CPs (dose: 200 mg·kg^−1^ body weight) daily.(iv).Medium dose collagen peptides group (MCPs): UV exposed + D-galactose; oral administration of 0.4 mL CPs (dose: 500 mg·kg^−1^ body weight) daily.(v).High dose collagen peptides group (HCPs): UV exposed + D-galactose; oral administration of 0.4 mL CPs (dose: 1000 mg·kg^−1^ body weight) daily.

D-galactose treatment was performed by the subcutaneous injection of 0.2 mL of 10% D-galactose solution at the back of the mouse neck (dose: 1.0 g/kg^−1^ body weight) daily. UV irradiation was performed with a 40 W UVA-340 LAMP (Q-panel, Cleveland, USA, peak wavelength: 340 nm), the distance between the lamp and the back of the mice was 30 cm (230 m J/cm^2^), and the irradiation lasted for 30 min every day for seven weeks (49 days). The radiation intensity was measured by a UVA365-radiometer (Xinbao Keyi Electronic Technology Co., Ltd., Xi’an, China). One hour after D-galactose and UV treatment, the mice were administered 0.4 mL of CPs orally every day. After the last irradiation, the mice were anesthetized, weighed, and the tissues were sampled for subsequent analysis.

### 2.5. Skin Moisture, Visceral Index, and Body Weight Gain

Skin moisture was determined by ISO 1442, and the visceral index was calculated using the following equation: visceral index (g/kg) = visceral weight/body weight.

### 2.6. Oxidative Stress, HA, and HYP Content of the Skin

Skin samples were homogenized with nine times the amount of normal saline (*w*/*w*) in an ice bath with a tissue homogenizer (TGrinder OSE-Y30, Tiangen Biochemical Technology Co., Ltd., Beijing, China), and then centrifuged at 2000× *g* and 4 °C for 10 min. The activities of superoxide dismutase (SOD), catalase (CAT), and glutathione peroxidase (GSH-PX), and the contents of hyaluronic acid (HA) and hydroxyproline (HYP) in the collected supernatant were determined according to the method described in the instructions corresponding to the respective kits.

### 2.7. Skin Histological

Mouse skin samples were fixed in 4% paraformaldehyde solution for 24 h, dehydrated, embedded in paraffin, and sliced. The skin sections were stained with hematoxylin and eosin (H-E) and observed with an ECLIPSE CI-E forward fluorescence microscope (Nikon, Japan). The thickness of skin epidermis and dermis was evaluated using the Halcon 13.0.1.1 software (MVTec, Munich, Germany).

### 2.8. Skin Transcriptome Sequencing

#### 2.8.1. RNA Extraction, Library Construction, and Transcriptome Sequencing

Total RNA was extracted from the skin of mice using the RNeasy Mini Kit (Tiangen Biochemical Technology Co., Ltd., Beijing, China) according to the manufacturer’s instructions. The purity and concentration of the RNA were checked using the kaiaoK5500^®^Spectrophotometer (Kaiao, Beijing, China); RNA integrity was assessed using the RNA Nano 6000 Assay Kit and the Bioanalyzer 2100 system (Agilent Technologies, Santa Clara, CA, USA). The transcriptional sequencing analysis of each sample was performed by Biolinker Technology Company Limited (Kunming, China). Briefly, the clustering of the index-coded samples was performed on a cBot cluster generation system using HiSeq PE Cluster Kit v4-cBot-HS (Illumina) according to the manufacturer’s instructions. After cluster generation, the libraries were sequenced on an Illumina platform, and 150 bp paired-end reads were generated.

#### 2.8.2. Bioinformatics Analyses of RNA-Sequencing Data

Gene Ontology (GO) and Kyoto Encyclopedia of Genes and Genomes (KEGG) enrichment analysis of differentially expressed genes (DEGs) was performed using the R language cluster analyzer package. When the *p*-value was less than 0.05, the items and pathways enriched by GO and KEGG were considered to be significant.

#### 2.8.3. Reverse Transcriptase-Polymerase Chain Reaction (qRT-PCR)

QRT-PCR was performed as previously described [19].

### 2.9. Western Blot

According to the method described by Park et al. [20], Western blot (WB) analysis was performed to quantify the expression of skin aging-related proteins in mice. The protein concentration of skin lysate in each treatment group was quantified using the BCA kit, separated by SDS-PAGE (10% acrylamide gel), transferred to a PVDF membrane, blocked with 5% skim milk, and incubated with an appropriate amount of primary antibodies (TGF-b1, c-Fos, c-Jun, Samd2/3, and β-actin) at 4 °C overnight. After washing with TBST, the samples were mixed with secondary antibodies and incubated at room temperature for 1 h. The ChemiDoc XRS + chemiluminescence gel imager (BioRAD, Hercules, CA, USA) was used to detect specific proteins. Image J software was used to quantify the expression of the target protein in each treatment group, and the results were represented by the density values normalized to β-actin protein.

### 2.10. ELISA

The expression levels of MMP-1, MMP-3, Type I pro-collagen, and IL-1α in skin lysis fluid were determined by enzyme-linked immunoassay. The assay was conducted according to the instructions provided with the kit.

### 2.11. Statistical Analyses

All results were analyzed using the SPSS 21 (IBM Inc., Armonk, NY, USA) software through one-way analysis of variance (ANOVA) and Duncan’s multiple range test, with the significance level set to *p* < 0.05. OriginPro 2017 (OriginLab, Northampton, MA, USA) was used to plot the data. All data were expressed as the mean ± standard deviation (SD).

## 3. Results and Discussion

### 3.1. Amino Acid Composition of Collagen

The amino acid composition of chicken bone collagen is shown in Table 1. Gly was the main amino acid in the samples, accounting for nearly one-third (27.86%) of the total amino acids, and Hyp was the special amino acid in collagen, accounting for 9.83%. The main characteristics of collagen molecules were the repeated Gly-X-Y sequences and the unique triple-helical structure composed of three α chains. Gly accounted for about one-third of the total amino acids, X and Y were often proline and hydroxyproline, or could be any residue [21]. The amino acid composition of the sample was similar to that of chicken bone collagen reported in previous studies and had the typical characteristics of collagen [22].

### 3.2. Molecular Weight Distribution of CPs

Molecular weight distribution reflects the degree of collagen hydrolysis. The molecular weight of the CPs was mainly below 3000 Da (Figure 1), accounting for about 87.61% of all collagen hydrolysates, indicating that almost all of the CPs were small peptides. Many studies claim that collagen peptides with a lower molecular weight have better biological activity [23]. For example, Song et al. [24]. reported that lower molecular weight (200–1000 Da, 65%) silver carp skin collagen peptides repaired UV-induced aging skin in mice more effectively than similar peptides with a higher molecular weight (>1000 Da, 72%). However, the German Gelita company showed through several clinical studies that the effect of collagen peptides is mainly determined by its matching degree with the properties of the collagen peptides after the degradation of human collagen, rather than the molecular weight of the collagen peptides. They found that the product VERISOL^®^ with an average molecular weight of 2000 Da has the most stimulating effect on skin fibroblasts, while the product Fortigel™ with an average molecular weight of 3000 Da has a special effect on cartilage repair [25,26,27].

### 3.3. Effect of Oral CPs on Alleviating Skin Aging

#### 3.3.1. Body Weight and Organ Index

The body-weight index and organ index are important and reflect the health status of mice. The weight of mice in each group increased normally during the test period (Table 2). The average weight gain of the M group was lower than that of the N group, and that of the CP treatment groups was higher than that of the M group, suggesting that CPs had no side effects on mice. In previous reports, the dose of collagen peptide-treated mice was mostly between 100–1000 mg/kg·bw·d, and the safe dose of tilapia collagen peptides reached 4.07 g/kg·bw·d [28,29,30]. Similarly, the growth of skin-aged mice after gavage with tilapia scale collagen peptides (dose: 500 and 1000 mg/kg·bw·d) was also similar to our results [29]. The spleen plays an important role in humoral and cellular immunity, thus, the spleen index is often used to evaluate immune system function. The liver index was used to evaluate the toxicity of the tested sample. In these tests, the liver and spleen indices in the M group were lower than those in the N group, and both recovered after treatment with CPs, but there was no significant difference across the treatment groups (*p* > 0.05). The results were similar to those of previous studies [30], indicating that CPs are safe and might slightly improve the immunity of mice.

#### 3.3.2. Skin Composition

The UV and D-galactose treatment (M group) significantly reduced the moisture, HA, and Hyp content in the skin, compared to those of the N group, by 13.36%, 24.08%, and 15.83%, respectively (*p* < 0.05) (Table 2). The contents of moisture, HA, and Hyp in the skin were significantly higher in mice after the oral administration of CPs compared to the contents of those in the mice of the M group (*p* < 0.05). Among the dose groups, the contents of the three skin components were significantly different between the LCP and HCP groups and were dependent on the dose of intake of CPs. The HCP group had even higher contents than the N group, and HA and Hyp were significantly different between the two groups (*p* < 0.05).

Changes in skin moisture content cause wrinkles and skin sagging and are affected by matrix components such as skin collagen and HA [31]. Hydroxyproline is a stable, rich, and special amino acid in collagen, and its content can indirectly reflect the content of collagen in the skin, as well as skin aging. Additionally, HA, which is highly expressed in skin ECM, plays an important role in controlling skin water balance, osmotic pressure, moisture retention, and elasticity as a water storage system and structural component of skin [32]. In this study, the content of moisture, HYP, and HA in the skin increased significantly after CP intake, which might be related to the promotion of collagen and HA synthesis by CPs. Moreover, the increase in HYP and HA increased the moisture content.

#### 3.3.3. Skin Histological Changes

The histological changes of the back skin of mice after seven weeks of continuous treatment are shown in Figure 2. The aging skin of the M group showed the characteristics of rough surface, thickened epidermis, thinned dermis, and sparse cells, compared to the skin of the N group. However, the condition of aging skin in mice improved after oral administration of CPs, maintaining a smooth, orderly, and complete structure similar to that in the mice of the N group. Thus, the skin dermis was significantly thinner, and the epidermis was significantly thicker in the M group than in the N group (*p* < 0.05). The change in skin dermis and epidermal thickness was significantly better after treatment with CPs (*p* < 0.05), and the effect was more obvious with the increase in the dose of CPs (Figure 2). The effect of oral CPs on the histological structure of aging skin was similar to that reported previously [4,28,31]. The increase in the thickness of the epidermis might be an adaptive change to protect the skin from external stimuli, loss of skin moisture, and UV damage, possibly due to the increase in UV-activated epidermal growth factor receptor (EGFR) that induces keratinocyte proliferation and epidermal hyperplasia [4]. However, the mechanism by which oral CPs alleviate the increase in epidermal thickness remains unclear. The dermis imparts elasticity and strength to the skin, and the degradation of ECM and the reduction in the ability to repair fibroblasts are the main causes of dermal thinning in aging skin. Dermal thickness increased after the treatment with CPs, which might be due to the removal of ROS from the skin and inhibition of the increase of MMPs by CPs. This, in turn, inhibited the degradation of skin collagen and elastin (Figure 3), the entry of CPs in the skin, and their participation in the synthesis of collagen and HA [6], which was confirmed by the increase in the content of Hyp and HA in the skin (Table 2).

#### 3.3.4. Skin Antioxidant and Inflammatory Levels

Determining the activity of antioxidant enzymes is the most commonly used method to evaluate the antioxidant level in the body [28]. As shown in Figure 3, the activities of CAT, SOD, GSH-Px, and MDA content in the M group were significantly lower than those in the N group (*p* < 0.05). Administering CPs effectively inhibited the decrease of CAT, SOD, GSH-Px activities, and the MDA content in the skin of mice, compared to those in the mice skin of the M group, and was positively correlated with the dose of CPs; there were significant differences among the different dose groups (*p* < 0.05). When ROS, accumulated by skin oxidative stress, exceed the antioxidant defense ability of the body, they destroy the ECM, which is the key cause of skin aging. ROS can cause the oxidation of lipids and proteins in the skin, resulting in fibroblast degeneration and changes in the skin. ROS activate the MAPK signaling pathway and the AP-1 protein to upregulate the expression of MMPs and promote the degradation of matrix collagen [21]. Although the antioxidant enzymes and antioxidants in the body can remove ROS to protect the skin from damage, when the content of ROS exceeds the defense (antioxidant) ability of the body or the body’s defense ability declines, it causes oxidative stress and skin aging.

Additionally, the cellular inflammatory response caused by ROS also contributes to skin aging. After UV and D-galactose treatment (M group), the content of IL-1α in the skin of the mice significantly increased compared to that in the N group (Figure 3D), indicating that the skin showed a significant inflammatory response (*p* < 0.05). The CPs significantly reduced the content of IL-1α in the skin in a dose-dependent manner, compared to that in the skin of mice in the M group. There was a significant difference between HCPs and LCPs (*p* < 0.05), indicating that CPs alleviated skin inflammation. The ^1^O_2_ produced by ultraviolet irradiation stimulated the expression of MMP-1 in dermal fibroblasts through the secretion of IL-1α and IL-6, thereby disrupting the ECM [33]. Therefore, this study suggested that CPs significantly increased the activity of skin antioxidant enzymes and inhibited inflammatory responses, which might be important in delaying skin aging in mice.

### 3.4. Mechanism of Action of Dietary CPs in Alleviating Skin Aging

#### 3.4.1. Analysis and Validation of RNA-Seq Data

Analysis techniques, such as PCA, HCA, gene GO enrichment, and KEGG pathway enrichment were used to analyze the transcriptome data. Based on the PCA analysis (Figure 4A) and hierarchical clustering analysis (heat map) of 4303 differential genes with average channel strength greater than 100, the relative expression levels of total DEGs between the two treatment groups are shown to provide an overview of the changes in gene expression (Figure 4B). The M group and the HCP group were significantly separated, and the expression patterns of most DEGs in the M and HCP groups were opposite, indicating that there were significant differences between the mouse skin after HCP treatment and the M group (Figure 4A,B). Pairwise comparisons showed that after feeding HCPs, 4303 genes were significantly expressed in the mice skin, including 1790 upregulated genes and 2513 downregulated genes (Foldchange > 2, *p* < 0.05) (Figure 4C). Among the six genes associated with skin aging quantified by qRT-PCR, five genes (including Fos, Jun, Cxcl1, and Egr1) were downregulated, and one gene was upregulated (Figure 4D), which was consistent with the overall trend of transcriptomic data. The RNA expression levels of Fos, Jun, and Cxcl1 were significantly upregulated in damaged skin compared to that in intact skin [34], and inhibition of Egr1 expression alleviated skin inflammation [35]. These results reflected the reliability of transcriptome sequencing data, and oral CPs effectively alleviated skin aging.

GO enrichment analysis and KEGG database enrichment analysis provided support for further investigating the biological functions of DEGs. GO analysis showed that DEGs were significantly enriched in biological processes such as DNA replication, regulation of hemopoiesis, regulation of hydrolase activity, and cytokine production; in cell components, which included lytic vacuole and lysosome, collagen-containing and other related genes were significantly enriched; in terms of molecular function, receptor-ligand activity, cytokine activity, and other related genes were significantly enriched (Figure 5). The KEGG enrichment analysis of the first 20 signal pathways significantly altered by DEGs is shown in Figure 5. Cytokine–receptor interaction, lysosome, neuroactive ligand–receptor interaction, cell cycle, systemic lupus erythematosus, and TGF-β pathway (*p* < 0.05) were closely related to aging, especially cytokine–receptor interactions, lysosomes, and TGF-β signaling. An important feature of cellular senescence is the accumulation of damaged organelles and protein aggregates. Lysosomes play an important role in degrading damaged organelles and protein aggregates in senescent cells [36]. Cytokine–receptor interaction is the main pathway of enrichment in the skin after being affected by various factors such as inflammation [37], sulfur mustard exposure [38], and terahertz pulse [39]. Cytokines, as small molecular proteins synthesized and secreted by various tissue cells, maintain skin homeostasis by controlling the balance between keratinocyte proliferation, differentiation, and apoptosis through complex interactions with growth factors [40]. The TGF-β signaling pathway is also important for regulating skin aging [14]. Gene functional enrichment analysis showed that TGF-β was highly expressed during cytokine–receptor interaction and in the TGF-β signaling pathway. TGF-β is a major pro-fibrotic cytokine that regulates cell differentiation and proliferation while inducing extracellular matrix protein synthesis [41]. Therefore, we verified the TGF-β signaling pathway and some cytokines. 

#### 3.4.2. Verification of the Mechanism of Action of CPs in Alleviating Skin Aging

To verify the role of the TGF-β signaling pathway and cytokine–receptor interaction in alleviating skin aging by oral CPs, WB analysis was performed to verify the TGF-β and transcription factor Smad2/3 in the TGF-β signaling pathway, as well as the key transcription factor AP-1 (c-Fos and c-Jun) that regulates cytokines. The contents of MMP-1, MMP-3, Type I pro-collagen, and IL-1α were determined by ELISA. The results of the WB analysis showed that the expression of TGF-β, Smad2, and Smad3 in group M was significantly lower than that in group N (*p* < 0.05). The expression levels of TGF-β and Smad3 were significantly higher in mice after the oral administration of CPs compared to their levels in group M mice; additionally, Smad2 also increased significantly (*p* < 0.05), except for in the LCPs in a dose-dependent manner (Figure 6). On the contrary, the expression of c-Fos and c-Jun in the M group was significantly higher than that in the N group. The expression of c-Fos and c-Jun in the CP group was significantly lower than that in the M group, except for the expression of c-Jun in the LCP group (*p* < 0.05), and the change was dose-dependent. These results indicated that the TGF-β signaling pathway was activated and AP-1 was inhibited after feeding CPs.

The AP-1 protein is a dimeric complex of Jun and Fos family proteins and is an important regulator of skin inflammation and cytokine expression. Generally, the complex composed of c-Jun and c-Fos shows the highest transcriptional activity in the skin [41,42]. ROS produced in aging skin cells first activate AP-1 protein and then regulate downstream cytokines (such as IL-1α), MMPs, and the TGF-β signaling pathway through transcription and translation, thus facilitating skin aging [43,44]. The TGF-β/Smad signaling reaction is a classical collagen synthesis pathway, and the Smad transcription factor is at the core of this signal transduction pathway. TGF-β triggers the phosphorylation and activation of downstream Smad2 and Smad3 by binding to the receptor, thereby increasing the synthesis of COLI [14].

Additionally, the ELISA results showed that the content of MMP-1, MMP-3, and IL-1α in the skin of the M group was significantly higher than that in the N group, and the content of Type I pro-collagen was significantly lower (*p* < 0.05). However, the contents of MMP-1, MMP-3, and IL-1α (Figure 3D) in the skin after oral administration of CPs were significantly lower than those in the M group (*p* < 0.05); Type I pro-collagen increased significantly, and all the changes were dose-dependent with CPs (Figure 7). MMPs are involved in the decomposition of skin collagen, IL-1α shows the level of inflammation of the skin, and Type I pro-collagen reflects the synthesis of skin collagen. Accumulating evidence suggests that the role of the Jun/AP-1 protein pathway has also been proposed to regulate skin inflammation [40]. The ELISA results showed that the combined treatment of UV and D-galactose caused skin collagen degradation, decreased collagen synthesis, and caused skin inflammation, leading to skin aging. However, these changes were reversed after the oral administration of CPs.

In addition to the above pathways, in this study, the up-regulated genes after CP treatment were significantly enriched into the lysosome pathway, indicating that CP treatment activated the lysosome in skin (Figure 5). Previous studies indicate that activated lysosomes clear aggregates and elevate the activation of senescent neural stem cells during aging [45]. In addition to these, the increase function of lysosome can reduce the concentration of intracellular ROS to prevent cell dormancy. Similarly, any decrease in the lysosome function can enhance intracellular ROS concentration that ultimately promote cell dormancy [46]. Although we have not systematically verified it in this paper, these results and previous reports imply that the activation of lysosomal function may be the main way for CPs to alleviate skin aging, and we will continue to aim to verify it.

Therefore, this study showed that dietary CPs could alleviate skin aging induced by UV and D-galactose in a dose-dependent manner. CPs alleviate skin aging by lowering skin oxidation level, inhibiting the expression of key transcription factors AP-1 (c-Jun and c-Fos), activating the TGF-β/Smad signaling pathway to promote collagen synthesis, inhibiting the expression of MMP-1 and MMP-3 (that inhibit collagen degradation), and inhibiting skin inflammation to alleviate skin aging (Figure 8). Moreover, our findings in the current study also suggest that the activated lysosome may be an important pathway for CPs to regulate skin aging, which deserves special attention.

## 4. Conclusions

In summary, this study confirmed that dietary supplementation of chicken bone collagen peptides could significantly alleviate skin aging induced by ultraviolet radiation and D-galactose through multiple pathways, which include promoting pro-collagen synthesis, inhibiting collagen degradation, improving skin antioxidant level, and inhibiting inflammation; the alleviation was dose-dependent with CPs. A detailed investigation showed that CPs first reduce the level of skin oxidation, inhibit the expression of the key transcription factor AP-1 (c-Jun and c-Fos), and then activate the TGF-β/Smad signaling pathway to promote collagen synthesis, inhibit the expression of MMP-1 and MMP-3 to inhibit collagen degradation, and inhibit skin inflammation to alleviate skin aging in mice. In addition, the activation of lysosomes may also be the main pathway for CPs to alleviate skin aging, which is worthy of follow-up research and verification. These results provide a theoretical basis for implementing collagen peptides to alleviate skin aging and broaden the scope for the comprehensive utilization of animal by-products in functional foods.

## Figures and Tables

**Figure 1 nutrients-14-01622-f001:**
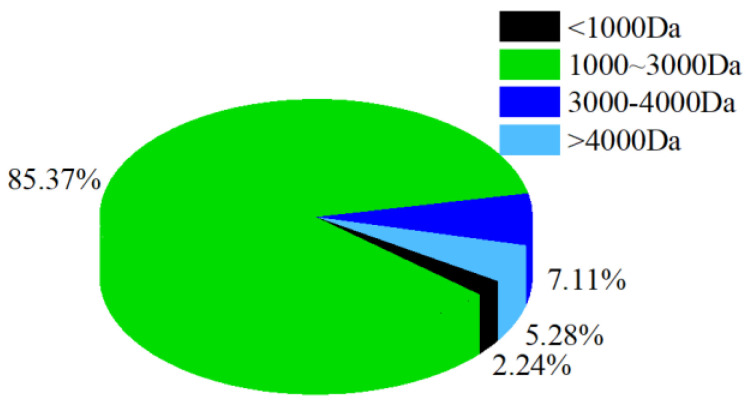
The molecular weight distribution of CPs.

**Figure 2 nutrients-14-01622-f002:**
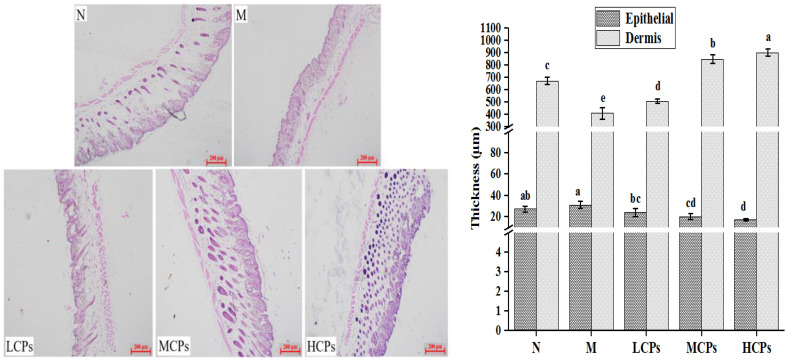
Effect of CPs on the histomorphology of aged skin (*n* = 3). Different letters in the same column indicate significant differences (*p* < 0.05), the same below. N—Normal group; M—Model group; LCPs—Low dose collagen peptide group; MCPs—Medium dose collagen peptide group; HCPs—High dose collagen peptide group. The same below.

**Figure 3 nutrients-14-01622-f003:**
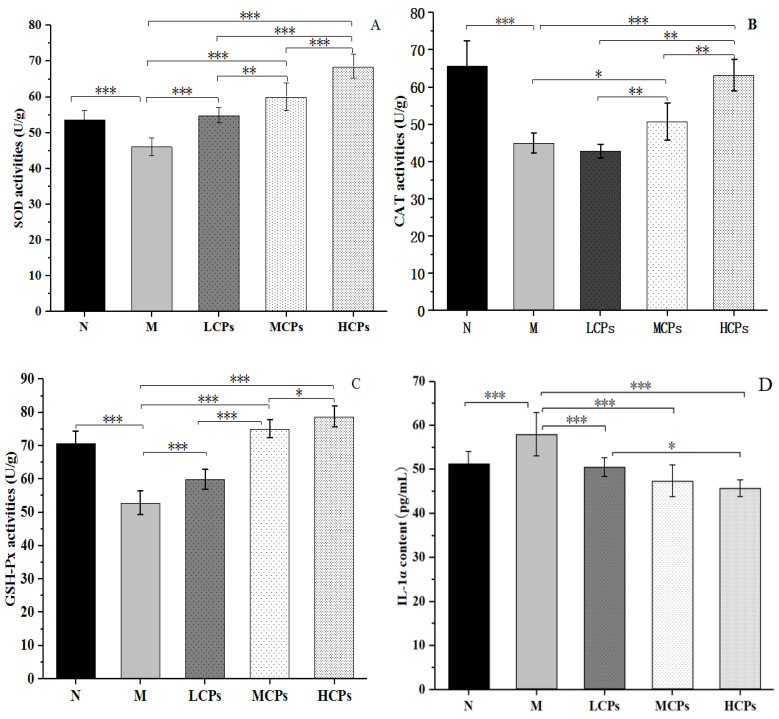
Effect of CPs on SOD (**A**), CAT (**B**), GSH-PX (**C**) activities and IL-1α content (**D**) in the skin (*n* = 10). Note: CP superoxide dismutase (SOD), glutathione peroxidase (GSH-PX), and catalase (CAT); *, ** and *** indicated that the data had statistically significant differences (*p* < 0.05), very significant (*p* < 0.01) and extremely significant (*p* < 0.001), respectively.

**Figure 4 nutrients-14-01622-f004:**
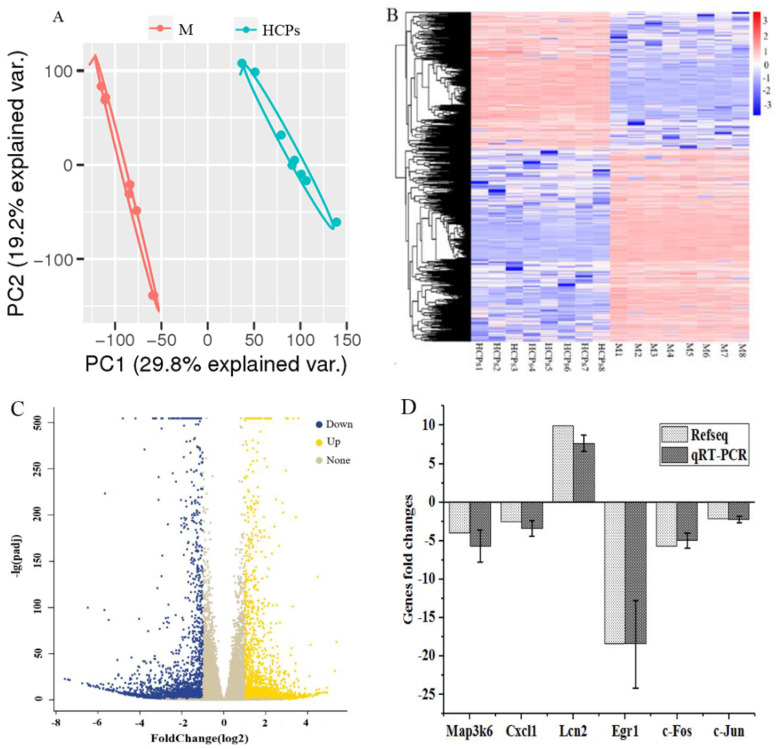
Differentially Expressed Genes (DEGs). (**A**) Volcano plot of DEGs; (**B**) Hierarchical cluster analysis of DEGs; (**C**) Volcano plot of DEGs; (**D**) Validation of the selected genes by qRT-PCR.

**Figure 5 nutrients-14-01622-f005:**
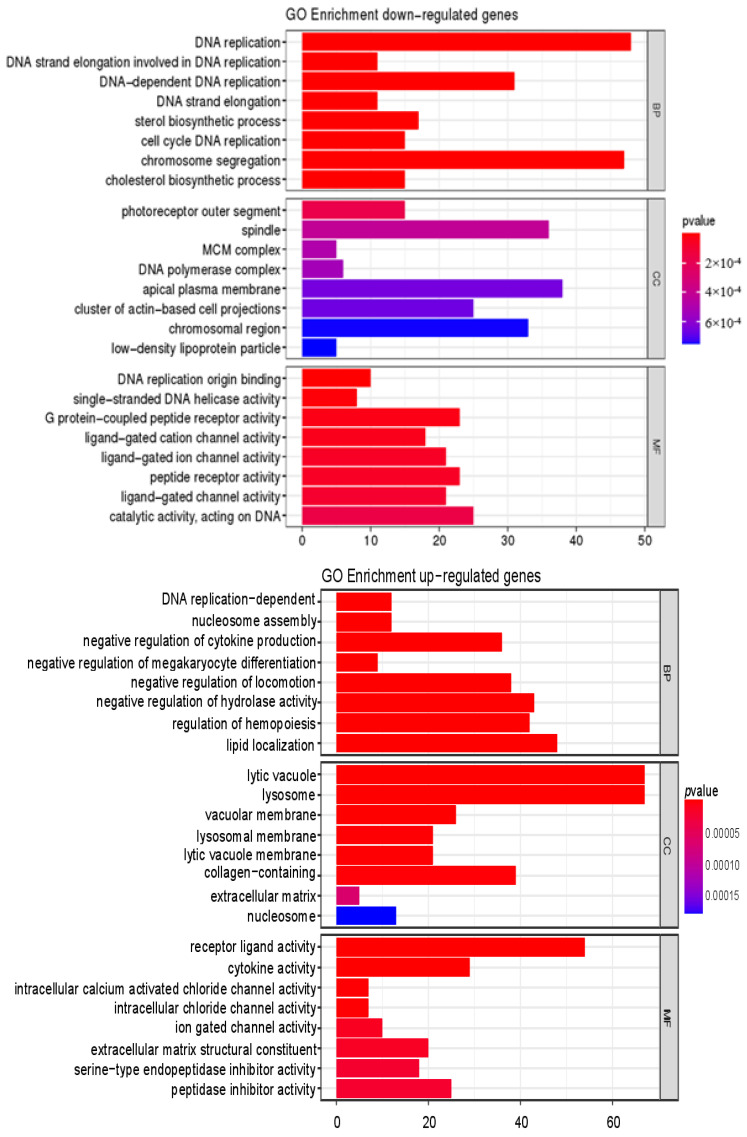
GO and KEGG functional enrichment analysis of DEGs in the skin.

**Figure 6 nutrients-14-01622-f006:**
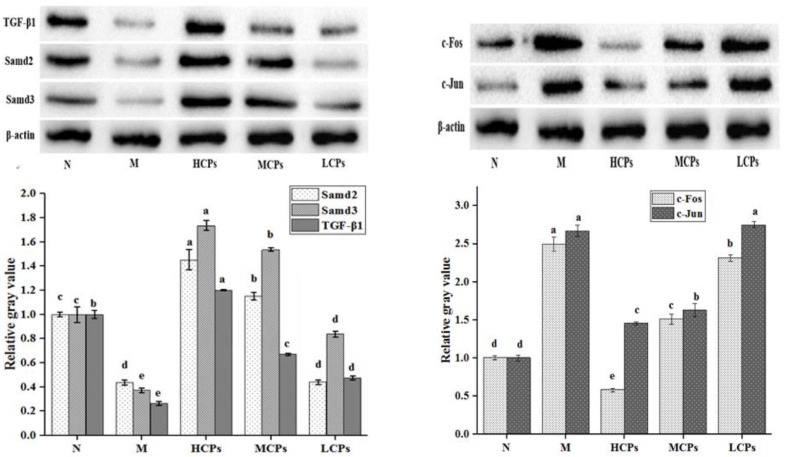
Effects of CPs on the TGF-B/Smad signaling pathway and AP-1 protein. Different letters in the same column indicate significant differences (*p* < 0.05).

**Figure 7 nutrients-14-01622-f007:**
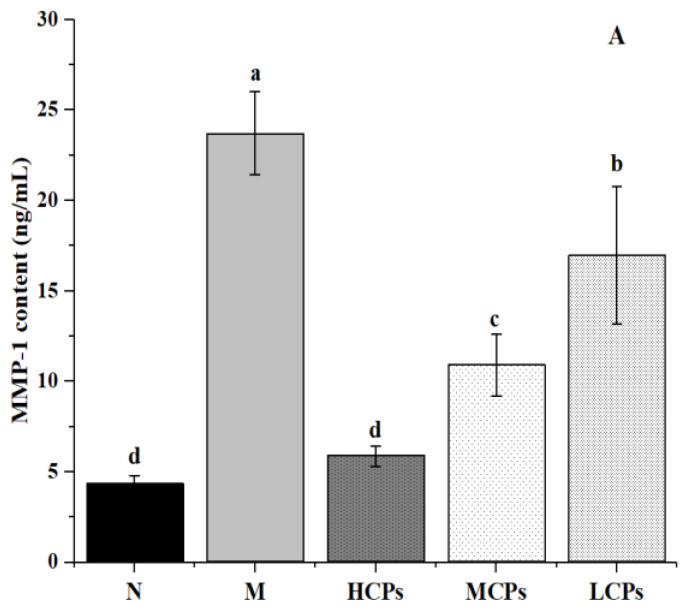
Effects of CPs on MMP-1 (**A**), MMP-3 (**B**), and Type I pro-collagen (**C**). Different letters in the same column indicate significant differences (*p* < 0.05).

**Figure 8 nutrients-14-01622-f008:**
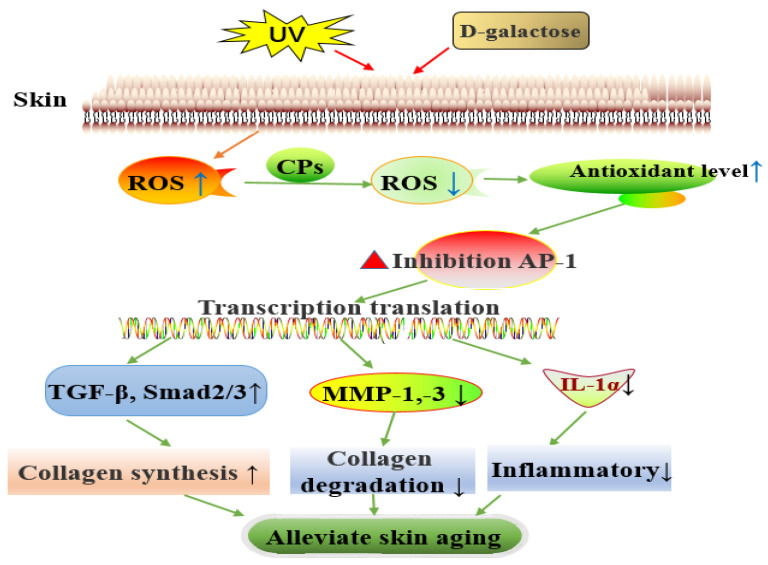
Biological pathways of dietary CPs to alleviate skin aging. “↑” means promote or increase; “↓” means inhibition or decrease.

**Table 1 nutrients-14-01622-t001:** The amino acid composition of chicken bone collagen.

Amino Acid Types	Content (%)	Amino Acid Types	Content (%)
Asp	4.45	Leu	2.54
Thr	2.37	Tyr	0.49
Ser	3.72	Phe	2.03
Glu	9.14	Lys	3.15
Gly	27.86	His	0.57
Ala	9.61	Arg	5.22
Val	1.45	Pro	11.60
Met	2.11	Hyp	9.83
Ile	1.48	Trp	0

**Table 2 nutrients-14-01622-t002:** Body-weight, organ index, and skin composition of mice.

Items	N	M	LCPs	MCPs	HCPs
Weight gain (g)	2.53	2.38	2.44	4.11	3.98
Moisture (%)	71.32 ± 1.61 ^ab^	61.79 ± 3.94 ^d^	68.78 ± 3.44 ^bc^	66.48 ± 4.40 ^c^	73.83 ± 1.88 ^a^
liver index (g/kg)	46.97 ± 3.03	43.09 ± 2.79	48.83 ± 3.64	49.91 ± 4.90	49.34 ± 5.48
Spleen index (g/kg)	4.86 ± 0.31	4.62 ± 0.52	5.53 ± 0.61	5.32 ± 0.68	5.55 ± 0.40
HA (pg/mg)	40.98 ± 2.18 ^b^	31.11 ± 2.70 ^d^	35.27 ± 1.54 ^c^	42.03 ± 2.26 ^ab^	44.47 ± 2.74 ^a^
Hyp (ug/g)	1531.19 ± 157.64 ^b^	1289.15 ± 153.15 ^d^	1399.37 ± 116.59 ^cd^	1639.26 ± 155.18 ^ab^	1749.79 ± 147.32 ^a^

Values are presented as mean ± SD (*n* = 10). Different letters in the same row indicate significant differences (*p* < 0.05), no letter indicates no significant difference (*p* > 0.05). N—Normal group; M—Model group; LCPs—Low dose collagen peptide group; MCPs—Medium dose collagen peptide group; HCPs—High dose collagen peptide group. The same below.

## Data Availability

The transcriptome raw data have been deposited in the NCBI website, and the BioProject number is: PRJNA783662.

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
