# Peer review of "Oral Intake of Chicken Bone Collagen Peptides Anti-Skin Aging in Mice by Regulating Collagen Degradation and Synthesis, Inhibiting Inflammation and Activating Lysosomes"

_nutrients, 2022, doi:10.3390/nu14081622_

Round 1
Reviewer 1 Report
Dear author of the manuscript "Oral intake of chicken bone collagen peptides anti-skin aging in mice by regulating collagen degradation and synthesis, inhibiting inflammation and activating lysosomes"
I have carefully read the manuscript. The study is well designed and results are well explained and discussed.
I have only one question. The dosages of CPs seem to be very high. As a example, converting the dosage to human ingestion, if the body weight of subject is 60 kg dosage of LCPs, MCPs, and HCPs will be 120g, 300g, and 600g respectively. As the converted amount, even LCPs is enough high dosage.
It may be necessary to explain this dosage point in discussion part.
Best regards.
Author Response
I have carefully read the manuscript. The study is well designed and results are well explained and discussed.
I have only one question. The dosages of CPs seem to be very high. As a example, converting the dosage to human ingestion, if the body weight of subject is 60 kg dosage of LCPs, MCPs, and HCPs will be 120g, 300g, and 600g respectively. As the converted amount, even LCPs is enough high dosage.
It may be necessary to explain this dosage point in discussion part.
Author's reply: Dear reviewer, thank you very much for your comments. The dose setting of CPs in the in vivo experiments in mice in this study was based on previous studies. Some studies reported that the safe dose of collagen peptides extracted from tilapia was 4.07 g/kg bw d (doi: 10.1016/j.jff. 2018.03.005), in fact, in previous reports, the treatment dose of collagen peptides was between 100-1000 g/kg bw d. We have explained this in the corresponding Discussion section. please check page 7, line 1, thank you again! Dosage reference literature such as:
- Oral Intake of Collagen Peptide Attenuates Ultraviolet B Irradiation-Induced Skin Dehydration In Vivo by Regulating Hyaluronic Acid Synthesis. doi:10.3390/ijms19113551
- Protective Effect of Gelatin and Gelatin Hydrolysate from Salmon Skin on UV Irradiation-Induced Photoaging of Mice Skin. doi: 10.1007/s11802-016-2953-5
- The effects and mechanism of collagen peptide and elastin peptide on skin aging induced by D-galactose combined with ultraviolet radiation. org/10.1016/j.jphotobiol.2020.111964
- Effects of collagen peptides intake on skin ageing and platelet release in chronologically aged mice revealed by cytokine array analysis. doi: 10.1111/jcmm.13317
- Effect of oral administration of collagen hydrolysates from Nile tilapia on the chronologically aged skin. doi:10.1016/j.jff.2018.03.005.
Reviewer 2 Report
General comments:
This paper offers a detailed investigation on the effect of collagen peptides from chicken bone in the skin aging process studied in mice. Methodology is sound, as well as the discussion. I have, however, some comments and suggestions, which I explain in detail below.
General formatting: Please, numbering of lines makes the reviewing process clearer and easier.
Specific comments:
Abstract line 2: I suggest modifying to -from marine organisms-;
Abstract line 3: Abbreviations should be presented the first time they appear in the text.
Abstract line 4: Enzymatically? With which enzyme? I suggest the authors briefly state the method of peptides preparation.
Keywords: Keywords should always be different from title to increase searchability. Moreover, “dose dependent” and “effect and mechanism” could be changed to more representative keywords.
Introduction:
- “ Inflamm-aging” ?
- Add a reference for the phrase “ in recent years…”.
- Between the previous paragraph and the paragraph that initiates with “ The wide application of collagen…” there could be a link between the use of collagen for skin anti-aging and then talk about collagen peptides. Why is collagen used for this purpose? Please add this info.
- What does it mean “complete peptides”?
- In the phrase “After 14 days, the skin still has a high content, …” High content of what? “Are almost be fully absorbed” is grammatically wrong and should be revised. Please revise the whole sentence.
- Where is the reference (s) for the phrase “religious beliefs…”.
Material and Methods:
- Correct the name of the section;
- Why are “glutathione peroxidase” and some other compounds in uppercase letters?
- How was the hydrolysate concentrated? There is no information whatsoever. After microfiltration? Was microfiltration used for concentration? Which membrane, conditions were used? Microfiltration is not used for the concentration of proteins and certainly not for peptides. Please explain.
- “According to the results of our previous optimization process”; where is the reference?? How can we consult these previous results?
- There is a repetition “ papain at an enzyme-to-substrate ratio…” Please revise.
- How was the papain enzyme inactivated?
- More details on the MS experiments should be given (solution in which the hydrolysate was dissolved, at which concentration, if cleaning kits were used, the concentration of protein on the analysed solution, etc.)
- Please write in full HA, HYP in the methods section, and for the enzymes as well.
- Statistical analyses: It is important to indicate if ANOVA assumptions were verified, and if so, by using which tests (example: test for normality check, Anderson Darling test). Test for homoscedasticity should also have been performed.
Results and Discussion
- There is an end-point missing at the end of the first paragraph.
- How many of those “peptides” below 3 kDa were actually amino acids? This is important information. The increase in activity reported for small peptides is true until a certain point. Amino acids are known not to express peptide-like activities.
- Please correct in the phrase ‘with a high molecular weight (bigger than 1000 Da)…” The correct word is “higher” and not “high” because peptides bigger than 1000 Da are not high molecular weight peptides.
- What was the standard error for the molecular weight distribution values? Add them in the Figure.
- Is there any toxicity study available on the toxicity of collagen peptides?
- Table 2: Why the standard deviations of weight measures are not there? How many times the analyses were done for each determination, 10 for all determinations? Please add this information to the methodology section. Also, difference test were not done for liver and spleen index or data is just not significantly different? If the last is the case, a comment on the table notes should be added.
- How many histological plates were analysed to generate those conclusions?
- Figure 2: I suggest separating Figure 2 (histological plates) and graph in two different figures. Also, adjust the graph because it is distorted.
- Graph on Figure 2: Does the effect of UV and galactose treatment explain the decrease in thickness observed for the dermis? What about the epithelial layer – does the radiation increase this layer thickness? Please add further comments on those effect – these results are quite interesting.
- Figure 3: Figures should be fully understood separately from the text. Thus, abbreviations should always be described in full in the Figure/table title.
- What would be the suggested dose of collagen peptides in mice for the increased health benefit?
Author Response
This paper offers a detailed investigation on the effect of collagen peptides from chicken bone in the skin aging process studied in mice. Methodology is sound, as well as the discussion. I have, however, some comments and suggestions, which I explain in detail below.
1.General formatting: Please, numbering of lines makes the reviewing process clearer and easier.
Author's reply: Dear reviewer, the line number and page number have been added to the revised manuscript, please check, thank you
Specific comments:
2.Abstract line 2: I suggest modifying to -from marine organisms-;
Author's reply: Dear reviewer, it has been changed to "from marine organisms" in the revised manuscript, please check, thank you
3.Abstract line 3: Abbreviations should be presented the first time they appear in the text.
Author's reply: Dear reviewer, the abbreviation "CPs" here refers to chicken bone collagen peptides, and the first CPs have been changed to "collagen peptides", which has been revised, thank you!
4.Abstract line 4: Enzymatically? With which enzyme? I suggest the authors briefly state the method of peptides preparation.
Author's reply: Dear reviewer, the enzymatic preparation of collagen peptides has been simply added to the abstract of the revised manuscript, please check, thank you
5.Keywords: Keywords should always be different from title to increase searchability. Moreover, “dose dependent” and “effect and mechanism” could be changed to more representative keywords.
Author's reply: Dear reviewer, thank you for your comments, we have changed "dose dependent" and "effect and mechanism" to "skin transcriptome" and "lysosome" respectively, which are our features, please Check, thank you!
Introduction:
6.“ Inflamm-aging” ?
Author's reply: Dear reviewer, "Inflamm-aging" is skin aging caused by skin inflammation induced by ROS accumulated inside the skin, and is one of the causes of skin aging, we have also reported it before, thank you for your comments !
7.Add a reference for the phrase “ in recent years…”.
Author's reply: Dear reviewer, thank you for your comments, references have been added, thank you!
8.Between the previous paragraph and the paragraph that initiates with “ The wide application of collagen…” there could be a link between the use of collagen for skin anti-aging and then talk about collagen peptides. Why is collagen used for this purpose? Please add this info.
Author's reply: Dear reviewer, thank you for your valuable comments, I have added the corresponding link in this section, please check, thank you!
9.What does it mean “complete peptides”?
Author's reply: Dear reviewer, thank you for your opinion. This should be absorbed and utilized in the form of "small peptide", which has been corrected to small peptide, please check, thank you!
10.In the phrase “After 14 days, the skin still has a high content, …” High content of what? “Are almost be fully absorbed” is grammatically wrong and should be revised. Please revise the whole sentence.
Author's reply: Dear reviewer, thank you for your comments, it has been carefully corrected, please check page 2, line 24, thank you!
11.Where is the reference (s) for the phrase “religious beliefs…”.
Author's reply: Dear reviewer, thank you for your comments, I have added the corresponding references, please check page 2, line 42, thank you!
Material and Methods:
Correct the name of the section;
12.Why are “glutathione peroxidase” and some other compounds in uppercase letters?
Author's reply: Dear reviewer, I have corrected it, please check page 3, line 8, thank you for your opinion!
13.How was the hydrolysate concentrated? There is no information whatsoever. After microfiltration?
Author's reply: Dear reviewer, I have clarified in the revised manuscript that nitrogen is used to dry and concentrate, please check page 3, line 35, thank you for your comments!
14.Was microfiltration used for concentration? Which membrane, conditions were used?
Author's reply: Dear reviewer, I have clarified in the revised manuscript that the collagen digestion solution is microfiltered with a 0.22 μm microporous membrane, please check page 3, line 36, thank you for your comments!
15.Microfiltration is not used for the concentration of proteins and certainly not for peptides. Please explain.
Author's reply: Dear reviewer, thank you for your comments, the "microfiltration" here refers to the microfiltration of protein digested liquid in the determination of protein amino acid composition, which has been explained in the revised manuscript, please check, thank you!
16.“According to the results of our previous optimization process”; where is the reference?? How can we consult these previous results?
Author's reply: Dear reviewer, thank you for your comments. Before preparing collagen peptides, we used the response surface method to optimize the optimal process conditions for enzymatic hydrolysis of collagen. Although we have not published this part yet, but We have listed the specific enzymatic hydrolysis conditions in this experiment. We believe that although the article has not been published yet, it will not affect the content, structure and conclusion of this article. Please consider, thank you for your suggestion!
17.There is a repetition “ papain at an enzyme-to-substrate ratio…” Please revise.
Author's reply: Dear reviewer, thank you for your comments, the duplicate content has been removed in the revised manuscript, please check, thank you!
18.How was the papain enzyme inactivated?
Author's reply: Dear reviewer, thank you for your comments. It has been stated in the revised manuscript that the enzyme is inactivated by boiling water bath for 15min. Please check page 3, line 41, thank you!
19.More details on the MS experiments should be given (solution in which the hydrolysate was dissolved, at which concentration, if cleaning kits were used, the concentration of protein on the analysed solution, etc.)
Author's reply: Dear reviewer, thank you for your comments. We have made changes in the revised manuscript. Please check, thanks!
20.Please write in full HA, HYP in the methods section, and for the enzymes as well.
Author's reply: Dear reviewers, I have added their full names in the revised manuscript, please check page 4, line 34, thank you!
21.Statistical analyses: It is important to indicate if ANOVA assumptions were verified, and if so, by using which tests (example: test for normality check, Anderson Darling test). Test for homoscedasticity should also have been performed.
Author's reply: Dear reviewer, I have to admire your knowledge for being comprehensive and professional. I'm sorry that we have been using SPSS software to perform variance analysis and Duncan's multiple test difference analysis on experimental data. But I will learn the relevant statistical knowledge you mentioned, and continue to improve my knowledge reserve in future research. Thank you very much for your opinion!
Results and Discussion
22.There is an end-point missing at the end of the first paragraph.
Author's reply: Dear reviewer, thank you for your comments, punctuation has been added to the revised manuscript, please check, thank you!
23.How many of those “peptides” below 3 kDa were actually amino acids? This is important information. The increase in activity reported for small peptides is true until a certain point. Amino acids are known not to express peptide-like activities.
Author's reply: Dear reviewer, thank you for your comments. Frankly speaking, we did not analyze the number of amino acids in small peptides, but the peptides <1000Da in these small peptides accounted for only 2.24%, and the molecular weight of amino acids was very low. The content of amino acids in collagen peptides is very small, so the number of amino acids in the peptides has very little effect on the activity of the experimental peptides, and in previous studies, the determination of the molecular weight distribution of peptides did not involve amino acids (DOI: 10.1039/c9fo01414d). But thank you for your opinion, we will study and improve it seriously in the future, thank you!
24.Please correct in the phrase ‘with a high molecular weight (bigger than 1000 Da)…” The correct word is “higher” and not “high” because peptides bigger than 1000 Da are not high molecular weight peptides.
Author's reply: Dear reviewer, thank you for your comments, it has been revised in the revised manuscript, please check page 6, line 11, thank you!
25.What was the standard error for the molecular weight distribution values? Add them in the Figure.
Is there any toxicity study available on the toxicity of collagen peptides?
Author's reply: Dear reviewer, thank you very much for your comments. Referring to previous research reports (DOI: 10.1039/c9fo01414d), the molecular weight distribution of collagen peptides is only an approximate analysis of the molecular weight range of peptides, so there is no standard error; in previous reports Collagen peptides are generally considered to be safe, and there are few reports on the toxicity of collagen peptides. Lin Wang et al. reported that the safe dose of collagen peptides extracted from tilapia by gavage in mice was 4.07 g/kg bw·d (doi: 10.1016/j.jff.2018.03.005). Thank you very much for your comments, we will continue to learn and improve in the follow-up research to solve these problems!
25.Table 2: Why the standard deviations of weight measures are not there? How many times the analyses were done for each determination, 10 for all determinations? Please add this information to the methodology section. Also, difference test were not done for liver and spleen index or data is just not significantly different? If the last is the case, a comment on the table notes should be added.
Author's reply: Dear reviewer, thank you for your valuable comments. The body weight of mice is indeed not weighed for each individual marker, but is weighed together for the average, so there is no standard deviation. In future experiments, we will Be careful. In addition, there is no significant difference in liver and spleen index, we have explained in the table note, please check page 7, line 32, thank you!
27.How many histological plates were analysed to generate those conclusions?
The author's reply: Dear reviewer, thank you for your comments. We have done a lot of HE-stained sections for each treatment group, and selected 3 of them to measure the thickness of the skin epidermis and dermis. It has been marked in the figure "n= 3", please check, thank you!
28.Figure 2: I suggest separating Figure 2 (histological plates) and graph in two different figures. Also, adjust the graph because it is distorted.
Author's reply: Dear reviewer, thank you for your comments. We think it is clear that Figure 2 is placed side by side, because the latter image is an intuitive reflection of the former image, and it cannot be accurately represented after separation. Of course, if the magazine's typesetting requires us Separate, we will separate as requested, thank you for your suggestion!
29.Graph on Figure 2: Does the effect of UV and galactose treatment explain the decrease in thickness observed for the dermis? What about the epithelial layer – does the radiation increase this layer thickness? Please add further comments on those effect – these results are quite interesting.
Author's reply: Dear reviewer, thank you for your comments. In fact, we have discussed the causes of thickening of the epidermis and thinning of the dermis in aging skin in 3.3.3, as follows “The increase in the thickness of the epidermis might be an adaptive change to protect the skin from external stimuli, loss of skin moisture, and UV damage, possibly due to the in-crease in UV-activated epidermal growth factor receptor (EGFR) that induces keratinocyte proliferation and epidermal hyperplasia [4]. However, the mechanism by which oral CPs alleviate the increase in epidermal thickness remains unclear. The dermis imparts elasticity and strength to the skin, and the degradation of ECM and the reduction in the ability to repair fibroblasts are the main causes of dermal thinning in aging skin” please check page 8, line 9-15, thank you!
30.Figure 3: Figures should be fully understood separately from the text. Thus, abbreviations should always be described in full in the Figure/table title.
Author's reply: Dear reviewer, thank you for your comments, we have revised according to your comments, please check page 9, line 19, thank you
- What would be the suggested dose of collagen peptides in mice for the increased health benefit?
Author's reply: Dear reviewer, thank you very much for your comments. The dose setting of CPs in the in vivo experiments in mice in this study was based on previous studies. Some studies reported that the safe dose of collagen peptides extracted from tilapia was 4.07 g/kg bw d (doi: 10.1016/j.jff. 2018.03.005), in fact, in previous reports, the treatment dose of collagen peptides was between 100-1000 g/kg bw d. We have explained this in the corresponding Discussion section. please check, thank you again! Dosage reference literature such as:
- Oral Intake of Collagen Peptide Attenuates Ultraviolet B Irradiation-Induced Skin Dehydration In Vivo by Regulating Hyaluronic Acid Synthesis. doi:10.3390/ijms19113551
- Protective Effect of Gelatin and Gelatin Hydrolysate from Salmon Skin on UV Irradiation-Induced Photoaging of Mice Skin. doi: 10.1007/s11802-016-2953-5
- The effects and mechanism of collagen peptide and elastin peptide on skin aging induced by D-galactose combined with ultraviolet radiation. org/10.1016/j.jphotobiol.2020.111964
- Effects of collagen peptides intake on skin ageing and platelet release in chronologically aged mice revealed by cytokine array analysis. doi: 10.1111/jcmm.13317
- Effect of oral administration of collagen hydrolysates from Nile tilapia on the chronologically aged skin. doi:10.1016/j.jff.2018.03.005.
